# Estimating Site-Specific Wind Speeds Using Gridded Data: A Comparison of Multiple Machine Learning Models

**Jintao Zhou** [1,2], **Jin Feng** [2,*], **Xin Zhou** [1,*], **Yang Li** [1] and **Fuxin Zhu** [2]

1 School of Atmospheric Sciences, Chengdu University of Information Technology, Chengdu 610103, China
2 Institute of Urban Meteorology (IUM), China Meteorological Administration (CMA), Beijing 100081, China
* Correspondence: jfeng@ium.cn (J.F.); zhouxin18@cuit.edu.cn (X.Z.)

**Abstract:** Accurate site-specific estimations of surface wind speeds (SWS) would greatly aid clean energy development. The quality of estimation depends on the method of interpolating gridded SWS data to derive the wind speed at a given location. This work uses multiple machine learning (ML) and deep learning (DL) methods to estimate wind speeds at locations across eastern China using the gridded fifth-generation data from the European Centre for Medium-Range Weather Forecasts. The root-mean-square error (RMSE) of these models' estimates for summer and winter are, respectively, reduced by 23% and 16% on average against simple linear interpolation. A deep convolution neural network (DCNN) consistently performs best among the considered models, reducing the RMSE by 26% and 23% for summer and winter data, respectively. We further examine the dependence of the models' estimations on altitude, land use category, and local mean SWS. And found that the DCNN can reflect the nonlinear relationships among these variables and SWS. Threfore, it can be used for site-specific wind speed estimates over a large area like eastern China.

**Keywords:** surface wind speed; gridded data; interpolation; machine learning

## 1. Introduction

The development of atmospheric models and data assimilation systems has made gridded meteorological data popular, as their wide geographical coverage and long timespans make them convenient for large-scale analyses. As gridded meteorological data represent regional averages within each grid, different horizontal resolutions would mean large differences between the gridded and actual data for a specific site. Current typical reanalysis datasets have a horizontal resolution of 0.25–1°, hindering their direct use in fine analyses.

Surface wind speed (SWS) is a variable that often needs site data that are more accurate than gridded values. For example, site assessment for a prospective wind farm needs accurate, long-term, in situ SWS data [1], outdoor sporting events such as the Winter Olympics are directly affected by the SWS at the location [2], and airports need the wind speed on the runway to ensure aircraft safety [3]. However, the complexities of ground conditions can make SWS vary sharply in the horizontal direction, resulting in significant differences between gridded SWS and in situ observations. Therefore, for a given location without in situ measurements, the gridded SWS data must be mapped to the location by one of a variety of methods. A uniformly applicable grid-to-site conversion scheme will greatly expand the applicability of gridded SWS data.

Interpolation is traditionally used for such conversion. There are various interpolation methods, but most of them, such as kriging, inverse distance weight interpolation, spline interpolation, etc., are mostly used to map discrete variables into a predicted surface [4]. Therefore, these methods are mostly used in meteorology for site-to-grid interpolation calculations. However, in this paper, we choose to interpolate from grid points to sites, so we choose two commonly linear methods (LM), nearest interpolation (NI) and bilinear

interpolation (BI) [5,6]. The former directly uses the values for the nearest grid point to a site, whereas the latter performs linear interpolation in each of the east–west and north–south directions. As these linear methods cannot consider factors such as terrain height or land use at a site. They also cannot employ a nonlinear relationship between the grid and site SWS data. Therefore, their estimation accuracy varies sharply grid-by-grid [5,7]. Regression models (RMs), including ridge regression and least absolute shrinkage and selection operator (LASSO) regression, partly overcome the constraints of linear interpolation because they can introduce estimating factors and nonlinearity by adding regularization terms [8,9]. However, such weak nonlinearity cannot reproduce the complex relationship between gridded and site SWS data. Previous studies have reported that linear interpolation and RMs often perform well in regions with generally homogeneous meteorological and land surface conditions but become worse in regions with complex land surface conditions [10,11]. Therefore, a grid-to-site conversion scheme for SWS is required to include more nonlinearities and various factors and to be widely applicable across a large region.

Machine learning (ML) has recently shown strong capabilities for handling nonlinear problems in a variety of fields [12,13]. Compared with traditional linear methods, ML can also conveniently introduce additional factors such as terrain height and land use; it is thus expected to give better interpolation results. For example, some studies have applied various ML models to interpolate data for earthquakes, solar radiation, and sea surface height [14–16]. Examples include ML-based missing values filling and resolution downscaling [17,18]. These prior studies all showed that ML improves the interpolation accuracy.

ML has also been used for SWS modeling, but mainly with a focus on forecasting including nowcasts [19–22], short-range forecasts [23–25], and correcting numerical weather prediction models' outputs for sites [26,27] and grid points [28,29]. Many variables related to SWS are incorporated into ML models to reduce SWS forecasting errors: these include meteorological data (e.g., surface pressure, air temperature, and humidity 2 m above ground level) and static data (e.g., terrain height and land use). Studies of ML-based grid-to-site conversion have focused mainly on small areas and been aimed at SWS data downscaling [30,31]. These studies have used ML methods to statistically downscale wind speeds over the UK, the Korean Peninsula in general, and South Korea specifically. Validation results have demonstrated that this method produces better results compared with any previous statistical method applied to wind resource assessment and that it is comparable to the results of dynamic downscaling.

Some unresolved problems with grid-to-site conversion remain. First, previous studies have considered a single site or small areas, rather than large areas [32]. Although some commonly used ML models, such as random forests (RF), extreme gradient boosting (XGBoost), and various deep learning (DL) models, perform well in small areas [33,34], there is a lack of studies applying them to large areas with complex ground surface conditions, such as China. Second, previous studies have mainly aimed to propose a given model rather than discuss the differences among various ML models, and thus they considered only a few models [35]. Therefore, it is necessary to systematically compare the commonly used ML models in a broad study. Third, owing to the significant differences in meteorological conditions, ML models should be trained separately for different seasons. For example, Liu et al. [29] established models based on RF for grid-to-site conversion in four seasons in the Beijing area and reported better performances in summer and autumn than in spring and winter. The potentially varying seasonal applicability is worth discussing for other commonly used ML models.

Eastern China is influenced by the East Asian monsoon and has continuous and stable wind resources. Therefore, there is high demand for SWS data at potential wind farm sites. We would like to discuss the usability of using machine learning methods for the uniform modeling of wind speed estimation over a large area such as eastern China by taking advantage of the fact that machine learning methods are computationally fast and can handle nonlinear problems. This study investigates grid-to-site conversion models for this

region using gridded data from the fifth-generation European Centre for Medium-Range Weather Forecasts global climate reanalysis dataset (ERA5) and 3 h measured wind speed at 10 m above ground level ($WS_{10}$) from the China National Meteorological Observation Network. We estimate the site $WS_{10}$ from ERA5 winter and summer data using various ML models, including decision tree (DT), RF, XGBoost, multilayer perceptron (MLP), and DCNN. We compare them with linear interpolation and two RMs, ridge and LASSO. The results show that the ML methods are significantly better than linear interpolation and the RMs, especially for summer data. Among the ML models, XGBoost and DCNN perform best. The effects of altitude, land use category (LUC), and mean $WS_{10}$ on the performance of these models are also discussed. We find that the DCNN outperforms the other models in estimating data for complex terrain and areas with high wind speed.

## 2. Data and Methods

### 2.1. Data and Samples

We use the ERA5 with a horizontal resolution of $0.25° \times 0.25°$ as the input gridded data. Six gridded surface meteorological variables are used to estimate $WS_{10}$ at the given locations: east–west wind ($U_{10}$), north–south wind ($V_{10}$), $WS_{10}$, air temperature ($T_2$), dew point temperature ($D_2$) at 2 m, and surface pressure ($P_0$). We also introduce two additional quasi-static variables, altitude (H) and LUC (Table S1), owing to their close relationship to $WS_{10}$ [36–40]. There are 30 LUCs. Unlike the other seven variables, LUC is a discrete variable labeled by number and cannot be input directly. Therefore, we use one-hot encoding to encode it into 30 LUC variables, each corresponding to one category. For an example grid cell with a LUC of water body, the water body variable has a value of 1, and the rest of the LUC variables have a value of 0. The target variable is the site-specific $WS_{10}$ observations over 2102 basic sites in eastern China (10–60° N, 105–152° E), and the time resolution of 3 h. The data are obtained from the China National Meteorological Observation Network (https://data.cma.cn/, accessed on 20 November 2021.) and are subject to strict quality control [37,41,42].

Linear interpolation uses only the $WS_{10}$ of the grid points near each site as input variables. The RMs and tree models (TM) have input variables that include all six gridded meteorological variables on a $5 \times 5$ grid around each site, the two static variables, and the LUC of the grid where the site is located. In addition, the distance between the site and each grid point is added as an input variable to introduce the relative position of the site in the grid. For the DCNN model, the input variables include all 38 variables on a $5 \times 5$ grid around each site given that the convolution can learn the spatial pattern around the sites.

### 2.2. Models

We compare nine grid-to-site conversion models for $WS_{10}$ data (Table 1). The LMs are simple NI and BI, which have shown similar performance for wind speed interpolation [6]. The traditional RMs are ridge regression and LASSO regression. Both models add a regularization term after the loss function of the linear regression, and thus their complexity can be effectively controlled to prevent overfitting [43]. Ridge and LASSO regression use $L_2$ and $L_1$ regularization, respectively. The former has less weight constraint than the latter [44,45]. Weight constraint allows a model to efficiently utilize spatial information, reduce the training time, and improve its estimation accuracy [8,9].

**Table 1.** Summary of models used in this study.

| Type | Linear Interpolation | Regression Models | Tree Models | Deep Learning Models |
|---|---|---|---|---|
| Name | Nearest Bilinear | Ridge Lasso | Decision Tree Random Forest XGboost | MLP DCNN |

The DT uses a bifurcation structure to fit the target value by traversing all features to determine the optimal decision option [46]. RF and XGBoost are based on DTs, but employ integrated algorithms, i.e., the final estimates are obtained by constructing multiple DTs and integrating all the results for voting. RF uses DTs that are independent of each other and are constructed by randomly sampling the dataset. The final output is obtained using a weighted average of the results obtained from all trees [47]. In the XGBoost models, the DTs are connected in series with each other. The later constructed DTs would fit the residuals of the previous tree to get the final output value by continuously reducing the residuals [48,49]. TMs are widely used for predicting wind speed and revising predictions [50,51].

The DL methods are the MLP and the DCNN. The MLP adds multiple hidden layers between the input and output layers and connects them with nonlinear activation functions [52]. The MLP used here has six hidden layers, the largest having up to 2048 hidden units, and the rectified linear unit (ReLU) as the activation function in each layer. Dropout [53] technology is used to increase the robustness of the model. The DCNN used here follows Zhu et al. [54]. This model is based mainly on DenseNet [55,56], but with the introduction of convolutional block attention modules (CBAM) [57]. Here we use the DCNN with three dense blocks. Each block is connected to the other two using CBAM and transition layers with the $1 \times 1$ convolution kernel.

We establish uniform modeling instead of point-by-point modeling for all seven nonlinear models. This is because unified modeling allows a model to be applied over a large range and quasi-static data (i.e., altitude and LUC) cannot be introduced into a point-by-point model as input variables given a fixed value of these variables at a specified location.In another word, regardless of whether the input grid data are dynamic or static, these models can convert them to wind speeds at the specified site at the corresponding time.

This work uses three metrics to assess the performance of the models. The root-mean-square error (RMSE), mean bias (MB), and the coefficient of determination ($R^2$). The two traditional RMs (Ridge and Lasso) and three TMs (DT, RF, XGBoost) were implemented on the Python platform using the scikit-learn library. The MLP and DCNN were implemented on the Python platform using the Pytorch library.

### 2.3. Training and Validation

All seven non-linear models use big data for training and validation. Considering the large sample size in this study, we use the leave-out method of validation, which directly divides the data into two mutually exclusive sets, one for training and the other for validation, to reduce the computation time. For summer data, we use values from June, July, and August 2019 and June 2020 for training and those from July 2020 for validation. The winter training data are from December 2019 and January, February, and December 2020; January 2021 data are for validation. The training and validation set contains about 2 million and 0.52 million samples, respectively, for both winter and summer (Table 2). Both RMs select the best regularization coefficients using 10-fold cross-validation. The three DT-based TMs have a maximum depth of 50 layers. The MLP and DCNN are trained over 20 epochs. All these models use the mean-square error as the loss function.

**Table 2.** Summary of data used in this study ("m" represents millions).

| Dataset | Training | | | | Validation | | Testing | |
|---|---|---|---|---|---|---|---|---|
| | Summer | | Winter | | Summer | Winter | Summer | Winter |
| Year | 2019 | 2020 | 2019 | 2020 | 2020 | 2021 | 2020 | 2021 |
| Month | 6, 7, 8 | 6 | 12 | 1, 2, 12 | 7 | 1 | 8 | 2 |
| Num. of times | 976 | | 976 | | 248 | 248 | 248 | 224 |
| Num. of samples | 2.05 m | | 2.05 m | | 0.52 m | 0.52 m | 0.52 m | 0.47 m |

Figures S1 and S2 show scatter plots for the seven models on the training and validation sets in the summer. Ridge regression has an $R^2$ of 0.43, MB of 0.02, and RMSE of 1.26 on the training set. LASSO regression has similar results on the training set ($R^2 = 0.42$, MB = 0.02, and RMSE = 1.27). The performances of both RMs with the validation set are slightly poorer than for the training set, but there is no significant overfitting: $R^2 = 0.38$, MB = 0.06, and RMSE = 1.24.

Compared with the two RMs, the three TMs have fewer errors. The DT has $R^2 = 0.51$, MB = 0.02, and RMSE = 1.17; XGBoost has $R^2 = 0.59$, MB = 0.2, and RMSE = 1.06. The results for DT and XGBoost are also consistent for the training and validation sets, indicating no significant overfitting. In contrast, RF performs very inconsistently on the training and test sets. The $R^2$ and RMSE on the training set are 0.92 and 0.46, respectively, and the corresponding values for the validation set are 0.54 and 1.06, respectively. This inconsistency indicates severe overfitting. Moreover, tuning the hyperparameter cannot reduce the overfitting.

The two DL models perform more consistently than the other models on the training and validation sets. The MLP has $R^2$ values of 0.45 and 0.45, MBs of −0.09 and −0.02, and RMSEs of 1.24 and 1.16, respectively, for the training and validation sets. DCNN shows the best and the most stable performance among all seven models, with $R^2$ values of 0.6 and 0.56, MBs of 0.03 and 0.1, and RMSEs of 1.05 and 1.04, respectively, for the training and validation sets.

Figures S3 and S4 show the seven models' results for the training and validation datasets for winter. The results are like those for summer: the DCNN outperforms the other models, and RF suffers from overfitting (albeit more severely than for the summer data). The overall RMSEs are slightly larger for winter than for summer. All models except the MLP show overestimation (positive MB) for the winter training set just like summer. For the validation set, all the models except the MLP overestimate $WS_{10}$ for summer but underestimate it in winter. This is attributable to the magnitude and variability of wind speed being greater in winter than in summer.

## 3. Results for the Test Dataset

### 3.1. Regional Averaged Estimation Error

The August 2020 (about 0.52 million samples) and February 2021 (about 0.47 million samples) datasets are the test sets for summer and winter, respectively (Table 2). Figures 1 and 2 show scatter plots of the observations and predictions for all the winter and summer test samples, respectively. For both winter and summer, the RMs have lower errors than the LMs, whereas the DL models and the TMs outperform the RMs. For summer, the DCNN has the highest predictive power among all schemes, with an $R^2$ value of 0.55 and RMSE of 1.09, better than ridge regression (the better regression scheme; $R^2 = 0.39$, RMSE = 1.27) and RF (the best TM; $R^2 = 0.54$, RMSE = 1.11). The results for winter and summer are similar, with the DCNN's $R^2$ of 0.64 and RMSE of 1.21 also outperforming the other schemes.

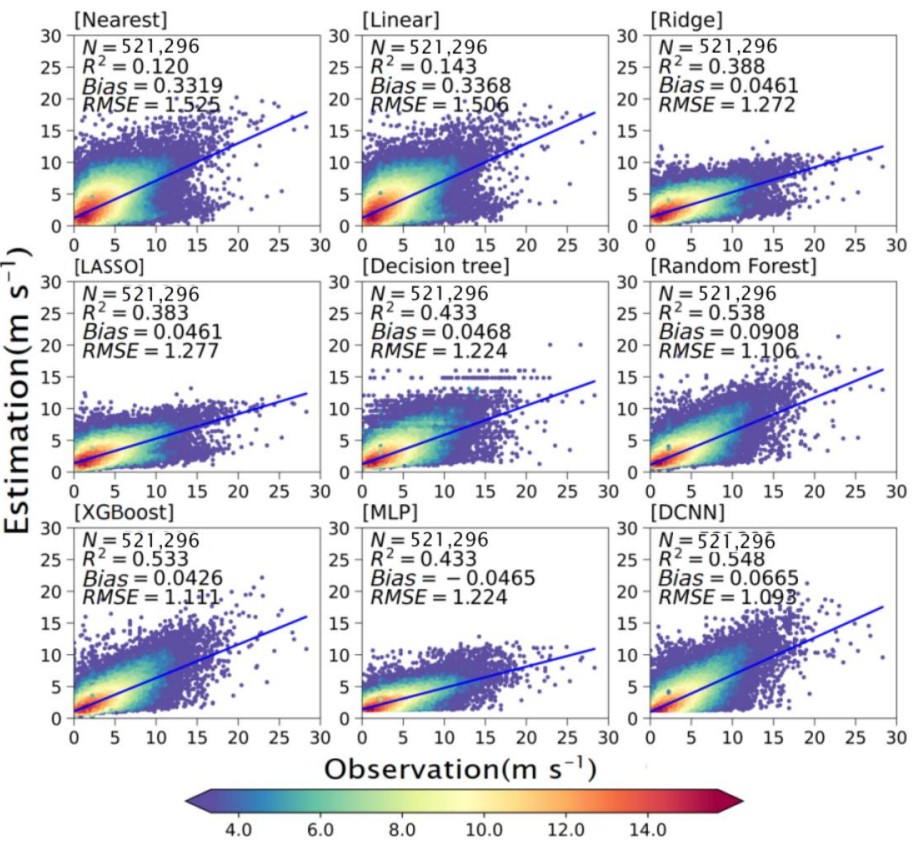

**Figure 1.** Scatter plots of wind speeds estimated by all nine grid-to-site conversion models and observed values for the summer test dataset. The value of colorbar is represented $\log_2$ *kernel density* $\times 100,000$.

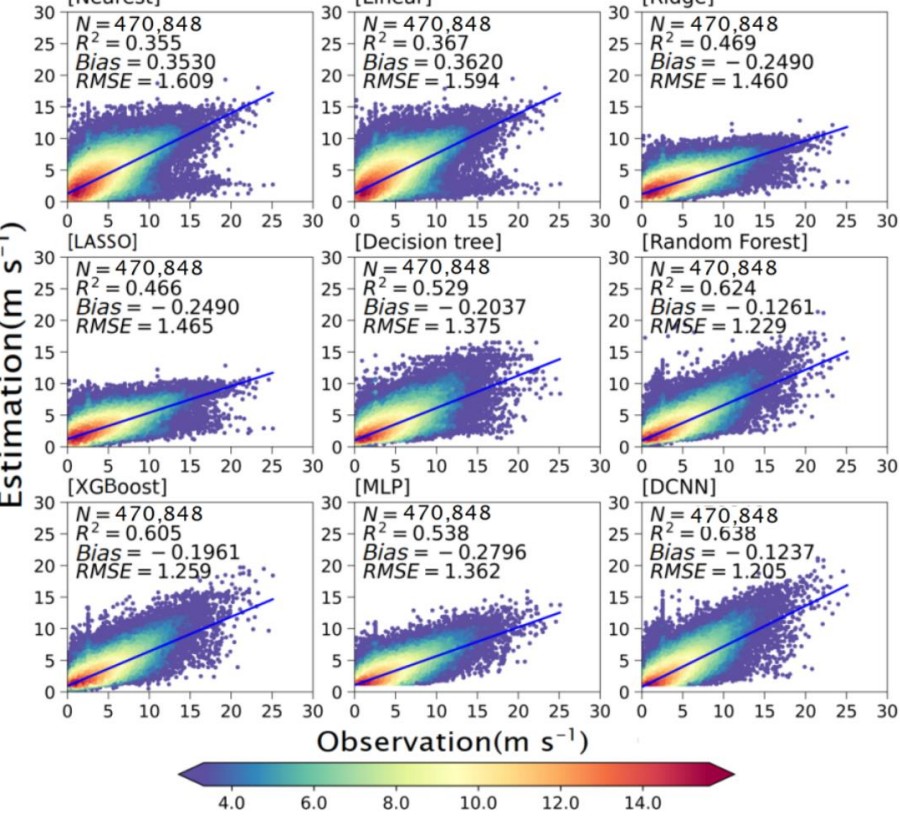

**Figure 2.** As Figure 1 but for the winter test dataset.

For summer, almost all models overestimate the $WS_{10}$ to varying degrees; only MLP underestimates $WS_{10}$. In terms of error amplitudes, all models have small errors (less than 0.1) except for the two traditional LMs, which significantly overestimate the $WS_{10}$ (nearest, MB = 0.33; linear, MB = 0.34). For winter, all the models except the LMs underestimate the $WS_{10}$. The DCNN performs slightly better (MB = $-0.12$) than the RMs (MB = $-0.25$) and RF (the best of the TMs; MB = $-0.13$).

Comparing each model's results for winter and summer shows that $R^2$ is higher for winter than for summer; however, RMSE shows the same trend. For example, DCNN has a better $R^2$ in winter than in summer (0.64 vs. 0.55) but a larger RMSE in winter (1.21 vs. 1.09). This is mainly attributed to the absolute $WS_{10}$ being greater in winter: the mean summer wind speed is 2.23 m s$^{-1}$, less than the winter average of 2.51 m s$^{-1}$.

To further characterize the distribution of RMSE, Figure 3 shows the probability density distributions (PDFs) of the RMSE of $WS_{10}$ in summer and winter for different models. In general, the means of the PDFs for both winter and summer are shifted to smaller values with smaller spreads for the ML methods in comparison with those for the conventional interpolation methods. For example, the mean RMSE for conventional interpolation is around 1.2 m s$^{-1}$, whereas the mean RMSEs for RM and DT are 0.9–1.0 m s$^{-1}$, and the XGBoost, RF, and DCNN models show even smaller values. The error distribution of these three models is also more concentrated.

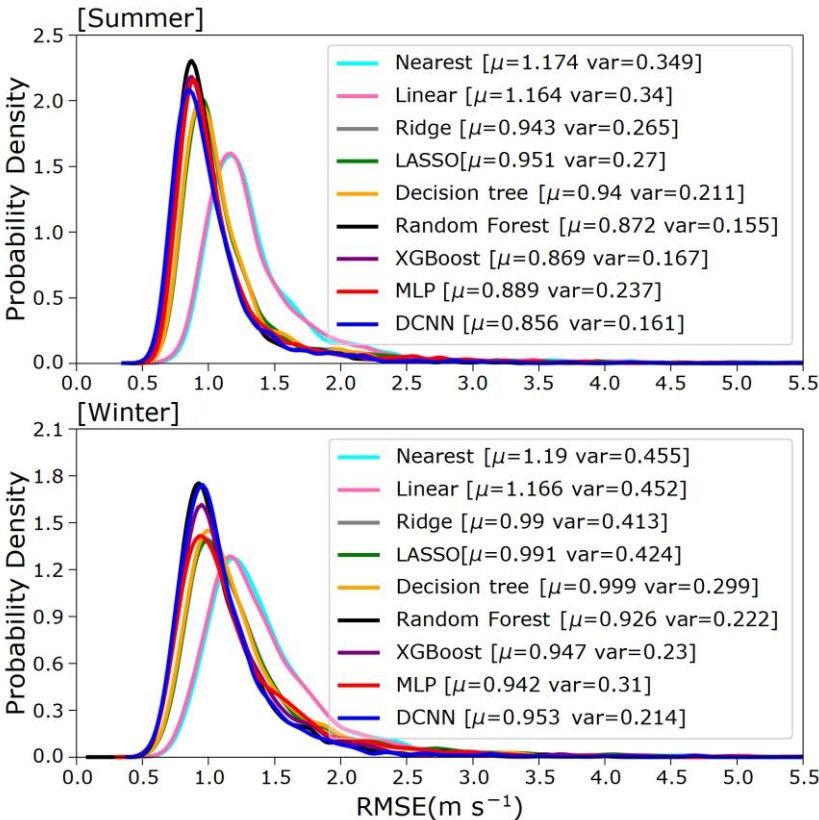

**Figure 3.** Probability density distribution of RMSEs of interpolation results for eastern China from different methods (μ: RMSE value corresponding to the peak, var: variance of all RMSEs).

### 3.2. Spatial Distribution of Estimation Errors

To further assess the performance of the models in different regions, we show the spatial distributions of the RMSEs of wind speeds estimated for summer (Figure 4) and winter (Figure 5) across eastern China. To quantify the effectiveness of estimation by these models in different regions, eastern China is divided into three regions using the 123° E line of longitude and the 32° N line of latitude (Figure 4): the three regions containing land are labeled South, Northeast (NE), and Northwest (NW). For the summer data, the

two LMs perform poorly overall, with similar error distributions. The regions with large errors are mainly concentrated in coastal areas, Inner Mongolia, and the Yunnan–Guizhou plateau. The RMSEs for the Yunnan–Guizhou plateau and Inner Mongolia are greater than 1.6 m s$^{-1}$. The RMSE for the NE (1.2–1.6 m s$^{-1}$) is better than that for Inner Mongolia and the Yunnan–Guizhou Plateau, but worse than that for Central China. The large errors for these regions indicate strong nonlinearities in the subgrid distribution of WS$_{10}$. This makes it difficult to resolve such errors by simple linear interpolation. Compared with simple interpolation, both RMs significantly reduce the RMSEs for site-specific WS$_{10}$ estimation. Compared with the poor performances of the LMs for Inner Mongolia, the Yunnan–Guizhou Plateau, and the NE, the RMSEs of the RMs are reduced by approximately 0.5, 0.2, and 0.2 m s$^{-1}$, respectively. In plains areas such as the NW, the RMSEs of the RMs are 0.6–1.0 m s$^{-1}$, significantly better than those of the LMs. The TM and the DL models yield better estimation results than the RMs, with RF, XGBoost, and the DCNN performing significantly better in almost all areas of eastern China. The spatial distributions of RMSE for winter are similar to those for summer, but with slightly larger values. For winter, the DCNN always has the smallest RMSEs among all the models for all three sub-regions (Table 3).

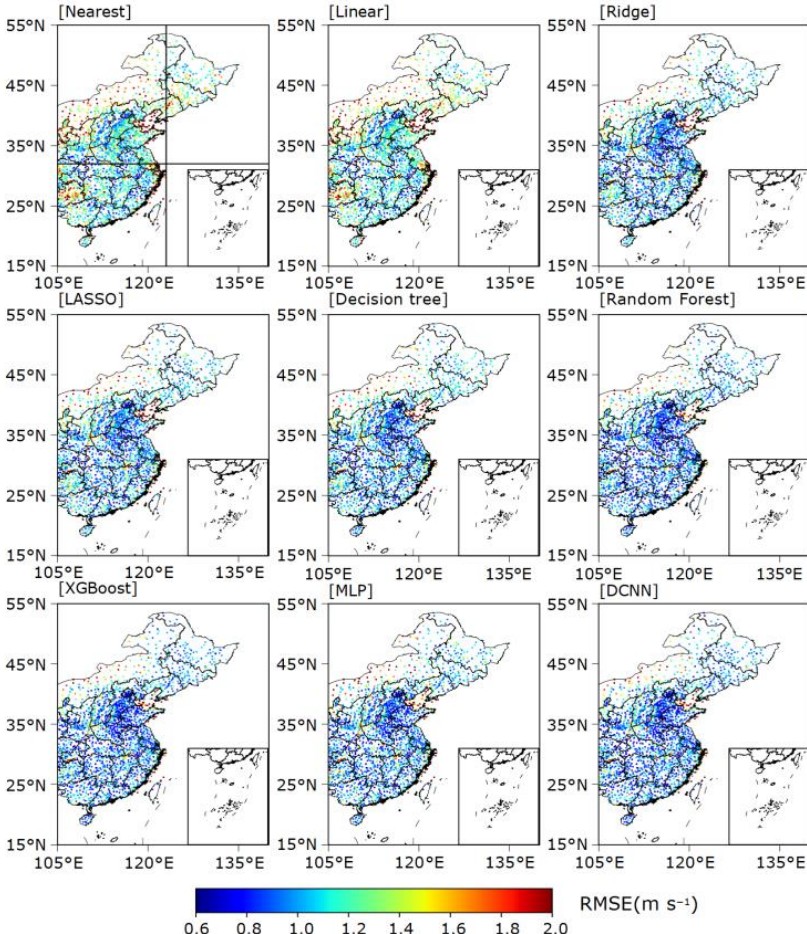

**Figure 4.** Spatial distribution of RMSEs of summer wind speeds across eastern China estimated by different interpolation methods.

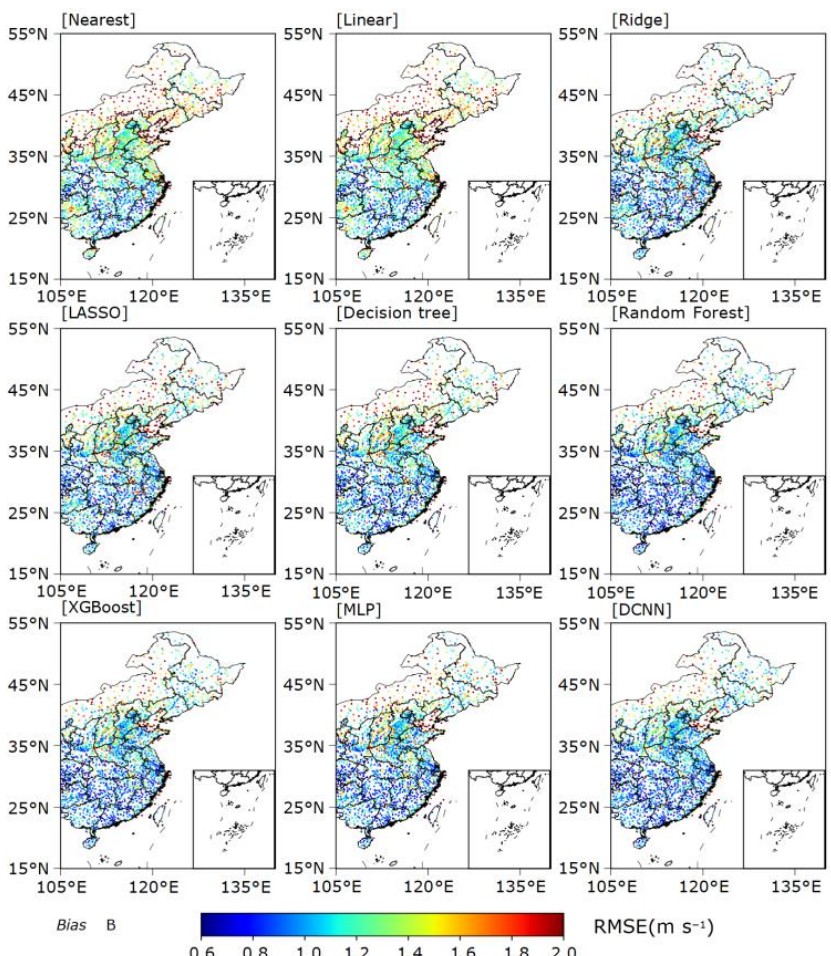

**Figure 5.** Spatial distribution of RMSEs of winter wind speeds across eastern China estimated by different interpolation methods.

**Table 3.** RMSEs and number of sites in three sub-regions of eastern China in summer and winter.

| Area | South China | | Northeast China | | North China | |
|------|--------|--------|--------|--------|--------|--------|
| | **Summer** | **Winter** | **Summer** | **Winter** | **Summer** | **Winter** |
| Nearest | 1.45 | 1.36 | 1.35 | 1.55 | 1.40 | 1.59 |
| Linear | 1.43 | 1.34 | 1.34 | 1.55 | 1.38 | 1.57 |
| Ridge | 1.20 | 1.20 | 1.10 | 1.40 | 1.14 | 1.43 |
| Lasso | 1.22 | 1.20 | 1.10 | 1.40 | 1.15 | 1.42 |
| Decision Tree | 1.16 | 1.12 | 1.10 | 1.33 | 1.14 | 1.40 |
| Random Forest | 1.06 | 1.02 | 1.00 | 1.20 | 1.05 | 1.26 |
| XGboost | 1.08 | 1.06 | 0.99 | 1.24 | 1.04 | 1.29 |
| MLP | 1.14 | 1.10 | 1.05 | 1.34 | 1.11 | 1.37 |
| DCNN | 1.04 | 1.02 | 0.99 | 1.18 | 1.03 | 1.24 |
| Num. of sites | 911 | | 1018 | | 173 | |

Figures 6 and 7 show the spatial distribution of $R^2$ values between the observed and estimated values for summer and winter. Similar to the distributions of RMSE, the $R^2$ values show that the RMs outperform the LMs but are worse than the ML or DL methods. For the summer data, the RMs show small $R^2$ values (0.15–0.3) for South China coastal provinces. The $R^2$ values for the TMs and DL models are above 0.2 for most sites; RF and

the DCNN show yet greater values, above 0.35. For the winter data, RF, XGBoost, and DCNN have $R^2$ values above 0.45, better than those of the LMs (0.2–0.4) and RMs (0.35–0.5). In addition, the $R^2$ of each model is higher for winter than for summer. This is related to the difference in wind speed between the two seasons.

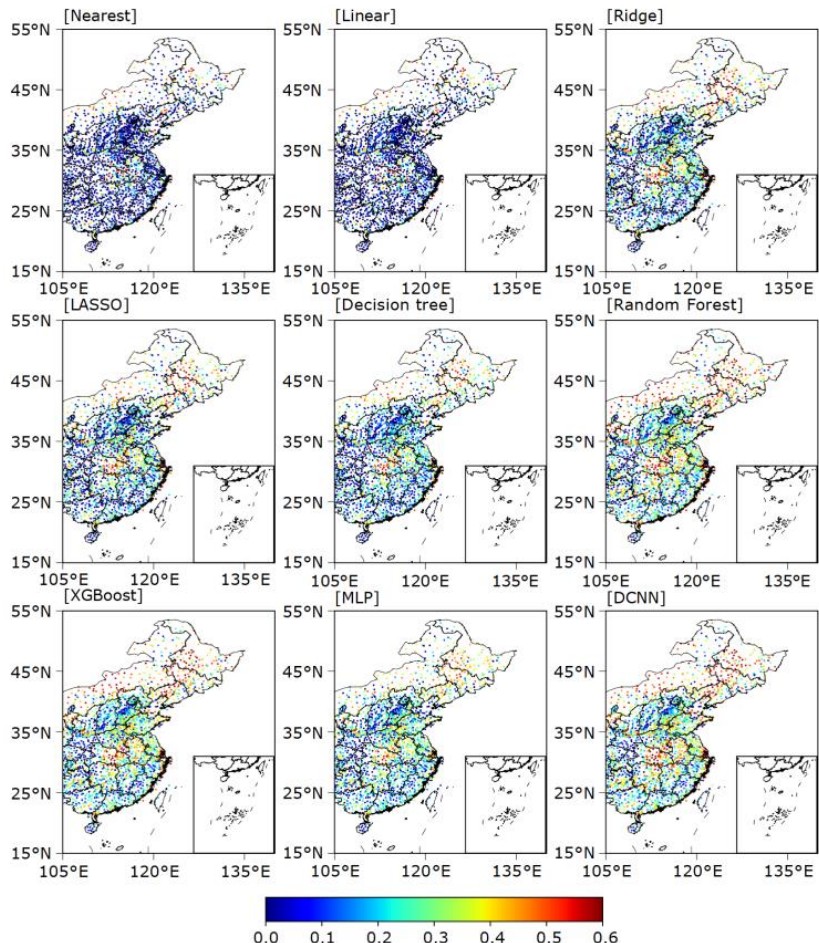

**Figure 6.** Spatial distribution of $R^2$ values for summer wind speeds across eastern China estimated by different interpolation methods.

To evaluate the accuracy of the estimated spatial distribution of $WS_{10}$, Figures 8 and 9 show scatter plots of observations and estimations for the time-averaged $WS_{10}$ at each site in summer and winter. For the summer, the RMs and TMs show RMSEs that are about 17% and 18% lower than those of the LMs; that of the DCNN, the best model, is 26% lower. Each ML and DL model shows a significantly larger $R^2$ value than the LMs and RMs: those of the LMs are all below 0.09, and those of the RMs are around 0.5. The $R^2$ values of the RF, XGBoost, and DCNN are 0.85, 0.86, and 0.85, respectively, which indicates that these ML and DL models can appropriately handle the spatial distribution of $WS_{10}$.

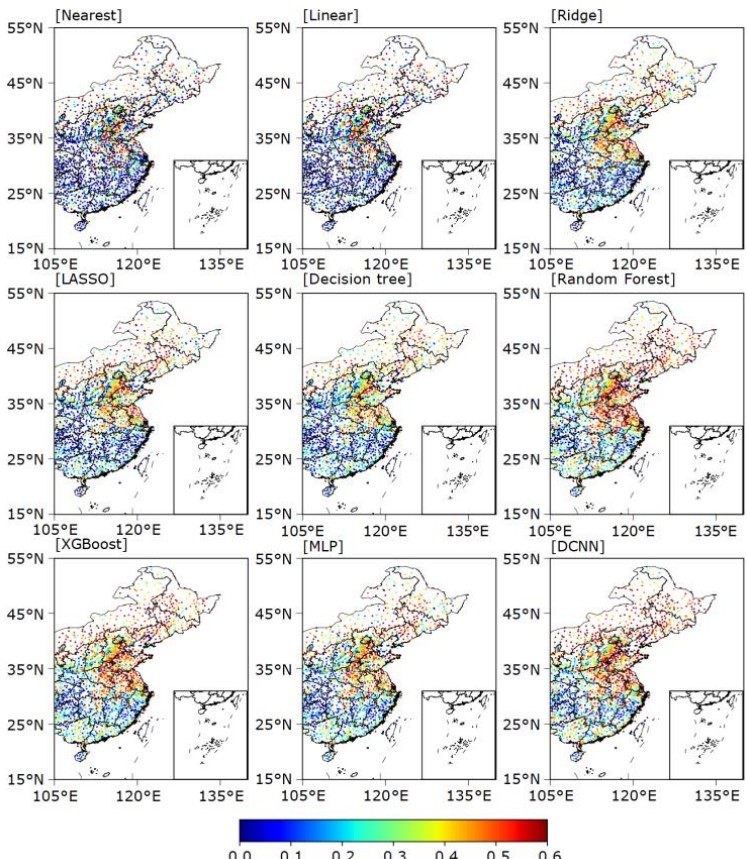

**Figure 7.** Spatial distribution of $R^2$ values for winter wind speeds across eastern China estimated by different interpolation methods.

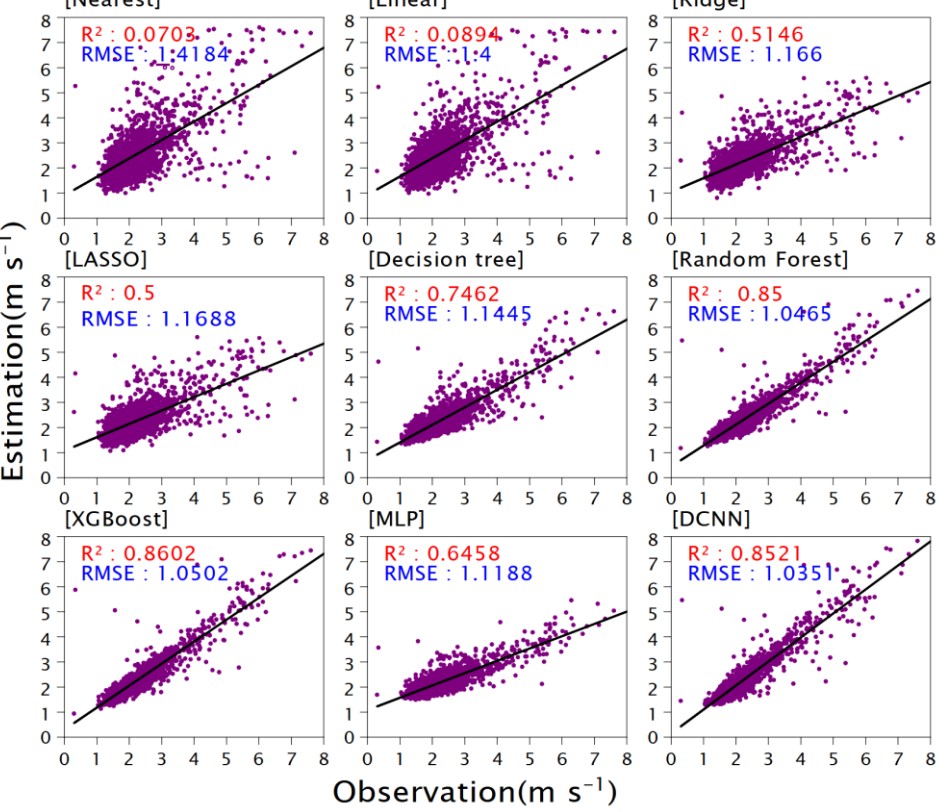

**Figure 8.** Correlation analysis of observed summer wind speeds and those predicted by each method.

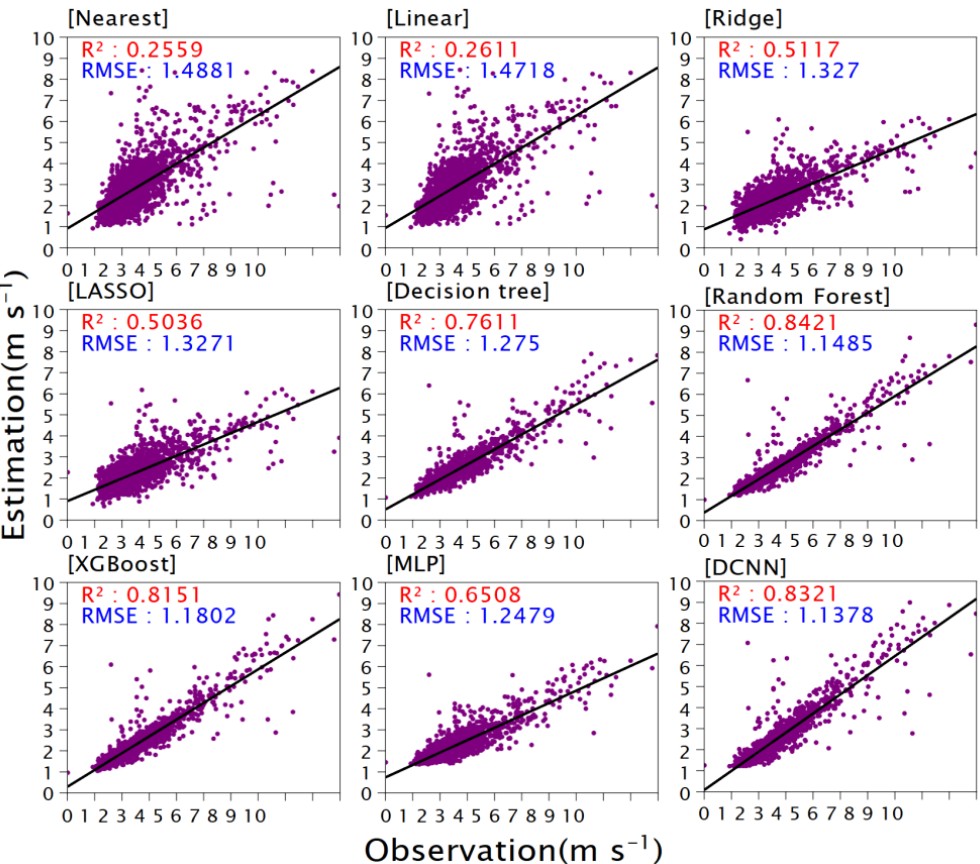

**Figure 9.** Correlation analysis of observed winter wind speeds and those predicted by each method.

For the winter data, the RMSEs of the RMs are at least 10.3% lower than those of the LMs. The reductions of the RMSEs relative to those of the LMs range from 13% for DT to 23% for the DCNN. The RF, XGBoost, and DCNN show the best performance: their respective $R^2$ values are 0.84, 0.82, and 0.83, similar to those for summer.

Overall, the RF, XGBoost, and DCNN outperform the other models for both winter and summer grid-to-site $WS_{10}$ conversion in eastern China. The DCNN performs best in most areas.

## 4. Dependence of Estimation Error on Altitude, LUC, and Mean $WS_{10}$

The ground condition significantly influences the $WS_{10}$. Previous studies have shown that errors in $WS_{10}$ estimation for China are strongly correlated with altitude, land use conditions, and $WS_{10}$ climate values [58]. Therefore, we analyze the dependence of estimation errors on different altitudes, LUC, and temporal mean $WS_{10}$ when using different grid-to-site $WS_{10}$ conversion models. The analyses illustrate the extent to which these various models can reflect the influence of these factors and thus assess the applicability of the models to different ground conditions.

### 4.1. Dependence of Estimation Error on Altitude

All the sites are grouped into 17 bins at 100 m intervals by elevation. The highest bin represents sites higher than 1500 m. Those with zero elevation are listed separately to show results for sites on the coast and on islands.

Figure 10 shows the averaged RMSEs for $WS_{10}$ in summer and winter for the altitude bins. The error in estimating $WS_{10}$ at zero elevation is the largest for both winter and summer. This may be related to the large errors in $WS_{10}$ values at sea level in the ERA5 datasets [59,60]. For summer, all models have the smallest RMSEs for sites at 200–300 m. For the sites with altitudes over 200 m, the RMSEs essentially increase with increasing

height. For the RMs, TMs, and DL models, the RMSEs do not vary significantly for the
0–500 m bins, as they all introduce the altitude as an estimation factor, but their RMSEs
increase with altitude for sites above 500 m. In contrast, the DCNN performs best for all
altitude bins and does not show significantly varying error with altitude. The results for
winter are similar to those in summer.

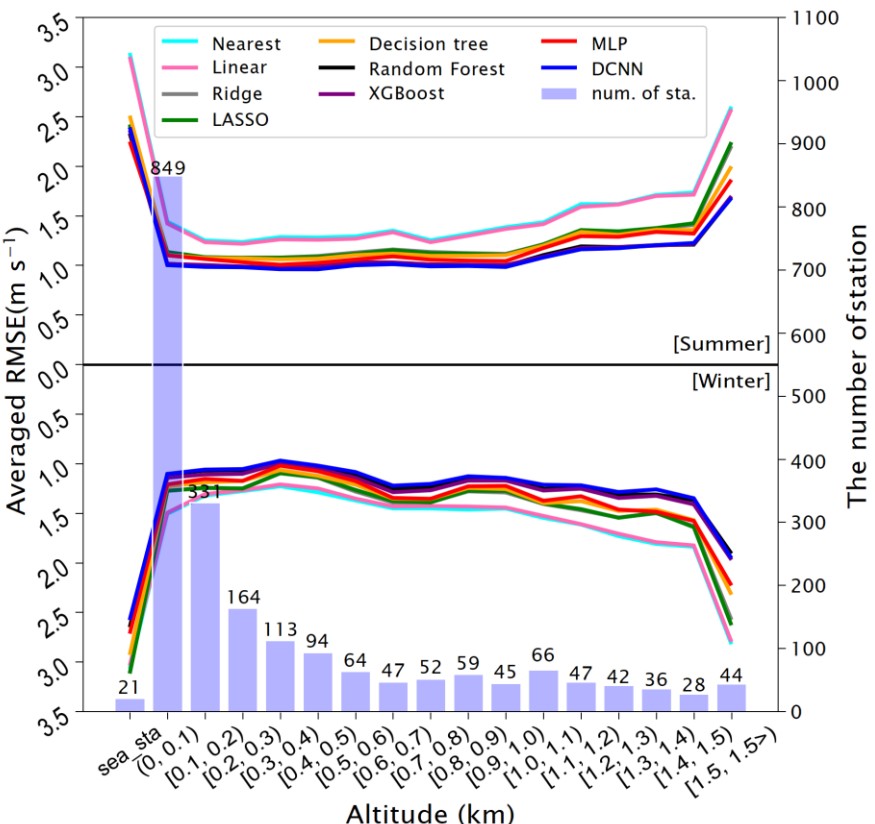

**Figure 10.** Comparison of errors in estimated wind speed at sites at different altitudes. Bars indicate
the number of sites at each altitude interval, and lines give the average RMSE of winter and summer
wind speeds estimated using each method at the corresponding altitude.

### 4.2. Dependence of Estimation Error on LUC

We classify all the LUCs into four types according to their typical roughness heights,
i.e., low-height surface (including cropland and grassland), tall vegetation (including
forests), urban, and water bodies. Figure 11 shows the reductions of RMSE (relative to
that of NI) for the various models applied to the winter and summer data considering
the four types of LUC. The BI performs similarly to NI for both seasons. For summer, the
estimations for low-height surfaces, water bodies, and urban areas are similar for all models:
the RMSEs of the RM and MLP are reduced by 0.2–0.3 m s$^{-1}$ in comparison with that of
NI. The RMSEs of RF, XGBoost, and DCNN are decreased by 0.3–0.5 m s$^{-1}$, mainly due to
the improved estimation of WS$_{10}$ over tall vegetation. The results for winter are similar to
those for summer. Overall, the DCNN gives the best estimations over all four LUC types.
RF performs slightly worse than the DCNN, but much better than the other models.

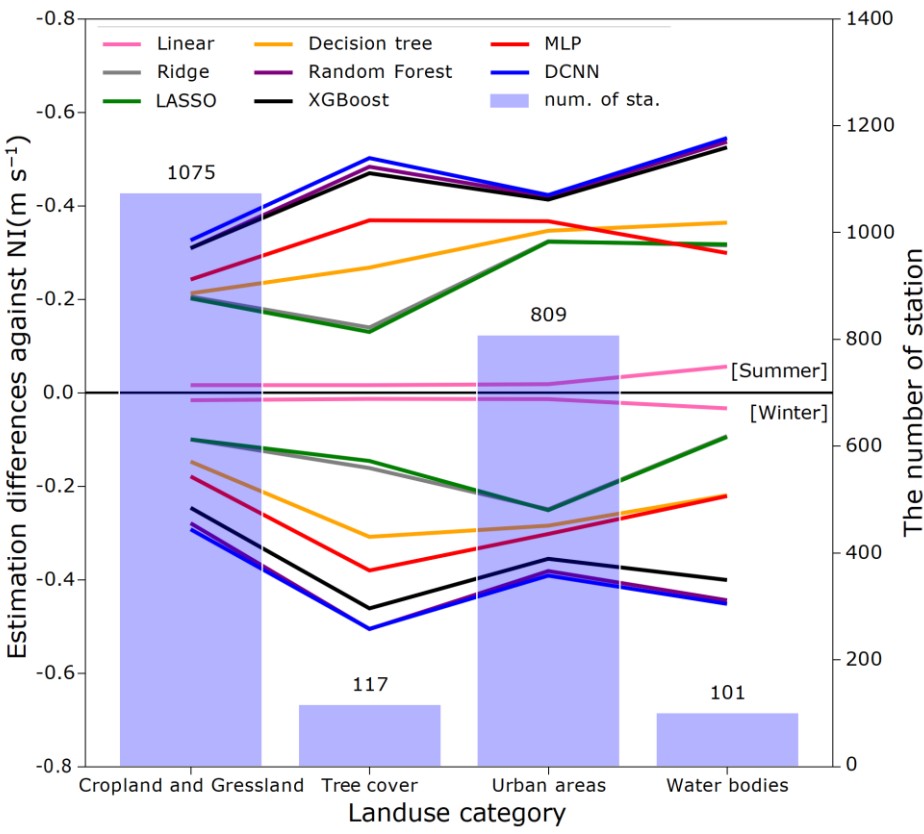

**Figure 11.** Comparison of errors in estimation of wind speed at sites with different land use types. Bars show the number of sites with each land use type, and lines compare the errors in winter and summer wind speeds estimated using each method at the corresponding altitude with the error of the estimation by linear interpolation. Note: the vertical Comparison method axis represents the difference in the error of the given method and that of NI (i.e., RMSE of the given method—the RMSE of linear NI). The land use types represent different underlying surface conditions, classified as arable and grassland (low vegetation), forest (high vegetation), urban areas, and water bodies.

*4.3. Dependence of Estimation Error on Mean $WS_{10}$*

　　Figure 12 shows the distribution of the estimated RMSE with respect to the mean wind speed. In areas where the mean $WS_{10}$ is between 0 and 2.5 m s$^{-1}$, the RMs, TMs, ML, and DL models perform similarly, much better than the LMs. In regions with mean $WS_{10}$ greater than 2.5 m s$^{-1}$, the RMSEs of both RMs and the MLP are close to those of the LMs, indicating that these models lose applicability to these areas. However, DCNN, XGBoost, and RF maintain their good estimation performance. The results for winter are similar to those for summer in areas with $WS_{10} \leq 2.5$ m s$^{-1}$. However, for areas with $WS_{10} > 2.5$ m s$^{-1}$, the RMs, MLP, and DT provide poor estimations, even worse than the two LMs. While XGBoost and RF perform better than those models, they are close to the two LMs in areas with 5.5–6.0 m s$^{-1}$ mean $WS_{10}$. Finally, the DCNN performs well in winter, with a significant advantage over the other eight models.

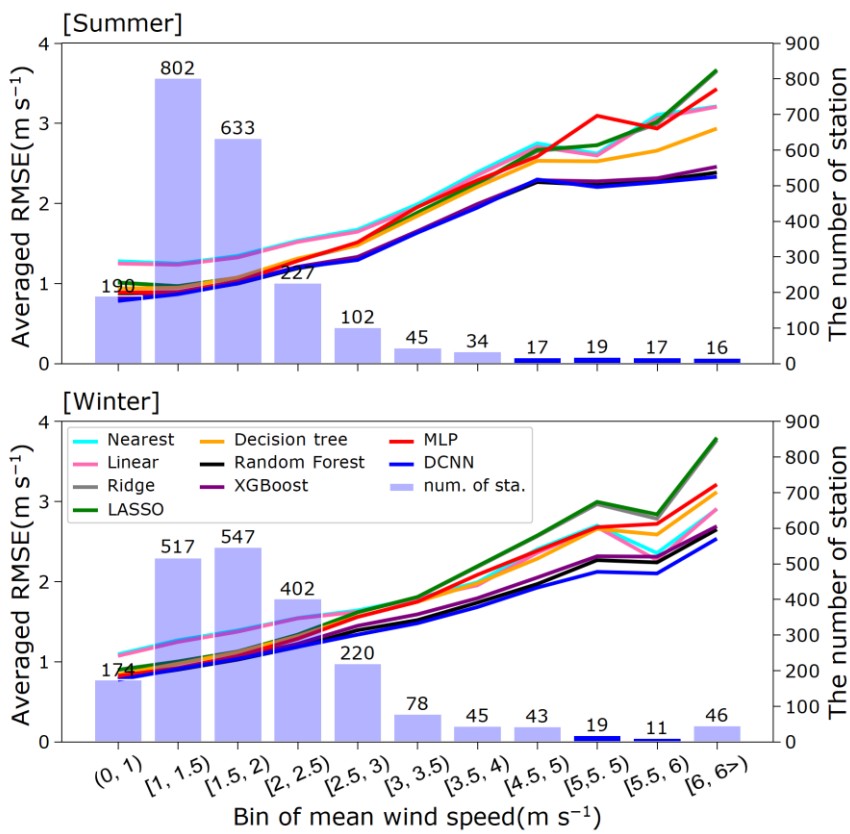

**Figure 12.** Comparison of errors in estimation of wind speed at sites with different average $WS_{10}$. Bars show the number of sites with $WS_{10}$ within each interval, and lines show the average RMSE of winter and summer wind speeds estimated using each method for the corresponding interval.

## 5. Conclusions

Interpolation is an important tool to fill gaps in gridded data to estimate $WS_{10}$ at a given sites for wind speeds. Traditional interpolation based on linear methods does not reflect the strong nonlinearity of the grid-to-site conversion of $WS_{10}$. Currently, models employing ML and DL are emerging as effective ways to deal with nonlinear interpolation problems. However, the effectiveness and applicability of various ML and DL methods become issues when applying a unified model to a large area like China. Therefore, we discuss the applicability of multiple ML and DL methods to estimate WS10 in winter and summer at sites across eastern China, and we compare these models with traditional LMs and RMs. The following conclusions are obtained:

(1) Overall, the estimation error of $WS_{10}$ is smaller for summer than for winter for all nine grid-to-site $WS_{10}$ models;

(2) The DT-based, ML, and DL models that use multiple input variables outperform the traditional LMs that use only gridded $WS_{10}$;

(3) Among these more elaborate models, the RF, XGBoost, and DCNN perform best;

(4) The DCNN is the overall best model as it performs robustly for sites at different altitudes and with the varying LUCs and local mean $WS_{10}$, indicating that it can reflect the nonlinear relationships among these variables and $WS_{10}$.

The main shortcoming of the ML and DL models for grid-to-site $WS_{10}$ conversion is that their performance varies significantly across regions, which limits their applicability as unified models. For example, none of these models provide satisfactory $WS_{10}$ estimations for northern Inner Mongolia, possibly owing to the lack of physical constraints of the surface layer in these unified data-based models. Similarly, surface wind speed interpolation in western China is not discussed in this paper because all the methods described above would lose validity in this region due to its complex climate, topography, and land condition.

A current proposal by Feng et al. [61] aims to develop a physics–data hybrid model that improves $WS_{10}$ estimation for almost all sites in China. However, the model does not apply any ML or DL structures, which would be detrimental to its handling of complex meteorology and ground conditions. Hence, the full integration of a physical model and ML or DL requires further study. In addition, this paper mainly calculates the site-specific $WS_{10}$ based on ERA5 reanalysis datasets. When these ML models are used for the grid-to-site conversion from data of numerical weather prediction, it is necessary to make an ML model with more input variables and a more complex structure to handle both the interpolation error and the error between forecasts and reanalyses.

**Supplementary Materials:** The following supporting information can be downloaded at: https://www.mdpi.com/article/10.3390/atmos14010142/s1, Figure S1: Scatterplot for all seven non-linear models on the summer training sets; Figure S2: Scatterplot for all seven non-linear models on the summer validation sets; Figure S3: Same as Figure S1 but for winter training sets; Figure S4: Same as Figure S2 but for winter validation sets; Table S1: Landuse value and label.

**Author Contributions:** J.Z.: Data Processing, Code Implementation, Literature Research, Result Visualization and Result Analysis, Writing—Original Draft. J.F.: Supervised, Conceptualization, Providing Resources, Writing—Original Draft, Writing—Review, and Editing. X.Z.: Supervised, Writing—Review, and Editing. Y.L.: Writing—Review. F.Z.: Provides design ideas for the DCNN model. All authors have read and agreed to the published version of the manuscript.

**Funding:** This research is supported by the National Key R&D Program of China (2021YFC3000901) and the National Natural Science Foundation of China (42275009, 41905037; 42275059).

**Institutional Review Board Statement:** Not applicable.

**Informed Consent Statement:** Not applicable.

**Data Availability Statement:** Data and code are available upon request from the corresponding author (Jin Feng).

**Conflicts of Interest:** The authors declare no competing interest.

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
