# Peer review of "Estimating Site-Specific Wind Speeds Using Gridded Data: A Comparison of Multiple Machine Learning Models"

_atmosphere, doi:10.3390/atmos14010142_

Round 1

Reviewer 1 Report

My comments are as below:
The abstract has to include the most important results from the author.

The introduction ought to have a clear hypothesis and set the stage for the next paragraph.

In general, there is excessive repetition throughout.

Figure ligands that were written in a careless manner.

This discussion has the potential to include additional significant information and sources.

Reorganize and go over the closing section with a rigorous editing pass.

Author Response

Responses to reviewer #1

for the manuscript entitled "Estimating site-specific wind speeds using gridded data: A comparison of multiple machine learning model" by Zhou et al.

Thank you very much for your comments and professional advice. These opinions help to improve academic rigor of our article. Based on your suggestion and request, we have revised the manuscript carefully, as described in our point-to-point responses to your comments.

  1. The abstract has to include the most important results from the author.

Response:

Done as your suggestion. Added all important points including the estimating results of multiple machine learning models and the advancetages of DCNN.

  1. The introduction ought to have a clear hypothesis and set the stage for the next paragraph.

Response:

Thanks for you suggestion. We added a sentence in the end paragraph of Section 1 to show our hypothesis and set the stage for the next contents. And put them in the attachment with red highlighting.

  1. In general, there is excessive repetition throughout.

Response:

Revised all duplicate texts throughout the manuscript.

  1. Figure ligands that were written in a careless manner.

Response:

We're sorry that our carelessness caused figure 6's mistake, and we have corrected it.

We have carefully revised the captions and legends of Figures 1-12.

  1. This discussion has the potential to include additional significant information and sources. Reorganize and go over the closing section with a rigorous editing pass.

Response:

In the last section, we added additional information and sources, including the geographic range to which the model can be applied. We also gave some discussion on applying the ML model to the grided data from numerical weather predictions. We have also revised the concluding section to make it easier for the reader to understand.

Reviewer 2 Report

The paper provided ML and DL estimation of site wind speed data. It was well written and organized. I recommend the publication with the following concerns.

1) Please further elaborate the practical use of the proposed method. Are they applicable for static data or dynamic data?

2) The advatage of the ML and DL method needs to be further discussed and compared. It seems to be better than linear interploation, which is the most basic method, what about method such as kriging, spline, etc.?

3) The software package used in this paper should be addressed.

Author Response

Responses to reviewer #2

for the manuscript entitled "Estimating site-specific wind speeds using gridded data: A comparison of multiple machine learning model" by Zhou et al.

Thank you very much for your comments and professional advice. These opinions help to improve academic rigor of our article. Based on your suggestion and request, we have revised the manuscript carefully, as described in our point-to-point responses to your comments.

  1. Please further elaborate the practical use of the proposed method. Are they applicable for static data or dynamic data?

Response:

All these proposed methods build uniform grid-to-site transformation models over all the eastern China grids and times. Thus, regardless of whether the input grid data are dynamic or static, these models can convert them to wind speeds at the specified site at the corresponding time. We added sentences in subsection 2.2 to elaborate the practical uses of these methods and put them in the attachment with red highlighting.

  1. The advantage of the ML and DL method needs to be further discussed and compared. It seems to be better than linear interploation, which is the most basic method, what about method such as kriging, spline, etc.?

Response:

The linear interpolation approaches using wind speed per se, i.e., bilinear interpolation and nearest interpolation, are the most basic methods. Decision tree should be a basic mothod for all tree models including random forest and XGBoost.

Kirging and spline methods are usually used for site-to-grid interpolation, while this study discusses grid-to-site interpolation. We have added sentences in Section 1 to explain the difference between these two methods and the methods used in this paper and put them in the attachment with red highlighting.

  1. The software package used in this paper should be addressed.

Response:

In this study, the two traditional regression models (Ridge and Lasso) and three tree models (Decision Tree, Random Forest, XGBoost) were implemented on the Python platform using the scikit-learn library. And the multilayer perceptron (MLP) and deep convolution neural network (DCNN) were implemented on the Python platform using the Pytorch library. I wrote the asdditions in section 2.2 and put them in the attachment with red highlighting.

Round 2

Reviewer 1 Report

Manuscript can be accepted for publication.